# Potential Antioxidant Multitherapy against Complications Occurring in Sepsis

**DOI:** 10.3390/biomedicines10123088

**Published:** 2022-12-01

**Authors:** Joaquin Abelli, Gabriel Méndez-Valdés, Francisca Gómez-Hevia, Maria Chiara Bragato, Silvia Chichiarelli, Luciano Saso, Ramón Rodrigo

**Affiliations:** 1Molecular and Clinical Pharmacology Program, Faculty of Medicine, Campus Norte, Institute of Biomedical Sciences, University of Chile, Avda. Independencia 1027, Santiago 8380000, Chile; 2Department of Biomedical Sciences, Humanitas University, Via Rita Levi Montalcini 4 , 20090 Pieve Emanuele, Italy; 3Department of Biochemical Sciences “A. Rossi-Fanelli”, Sapienza University of Rome, 00185 Rome, Italy; 4Department of Physiology and Pharmacology “Vittorio Erspamer”, Faculty of Pharmacy and Medicine Sapienza University, P.le Aldo Moro 5, 00185 Rome, Italy

**Keywords:** oxidative stress, sepsis, septic shock, antioxidants, multitherapies

## Abstract

Septic shock currently represents one of the main causes of mortality in critical patient units with an increase in its incidence in recent years, and it is also associated with a high burden of morbidity in surviving patients. Within the pathogenesis of sepsis, oxidative stress plays an important role. The excessive formation of reactive oxygen species (ROS) leads to mitochondrial damage and vasomotor dysfunction that characterizes those patients who fall into septic shock. Currently, despite numerous studies carried out in patients with septic shock of different causes, effective therapies have not yet been developed to reduce the morbidity and mortality associated with this pathology. Despite the contribution of ROS in the pathophysiology of sepsis and septic shock, most studies performed in humans, with antioxidant monotherapies, have not resulted in promising data. Nevertheless, some interventions with compounds such as ascorbate, N-acetylcysteine, and selenium would have a positive effect in reducing the morbidity and mortality associated with this pathology. However, more studies are required to demonstrate the efficacy of these therapies. Taking into account the multifactorial features of the pathophysiology of sepsis, we put forward the hypothesis that a supplementation based on the association of more than one antioxidant compound should result in a synergistic or additive effect, thus improving the beneficial effects of each of them alone, potentially serving as a pharmacological adjunct resource to standard therapy to reduce sepsis complications. Therefore, in this review, it is proposed that the use of combined antioxidant therapies could lead to a better clinical outcome of patients with sepsis or septic shock, given the relevance of oxidative stress in the pathogenesis of this multi-organ dysfunction.

## 1. Introduction 

According to the “Third International Consensus Definitions for Sepsis and Septic Shock” (Sepsis-3), sepsis is a clinical syndrome associated with life-threatening organ dysfunction caused by a dysregulated host response to infection [1]. In recent years, sepsis has presented an increase in its incidence, probably due to the increase in life expectancy, a higher prevalence of comorbidity, and due to the changes in definition that have existed as this pathology is more studied [2]. It affects between 19 and 48.9 million people a year and continues to be one of the main causes of morbidity and mortality in the world [2]. A subset of sepsis patients will develop or initially present with septic shock (SS), in which there are deep circulatory, cellular, or metabolic abnormalities profound enough to increase the risk of mortality [3]. Sepsis survivors have a twofold increased cardiovascular risk compared to the normal population, which is reflected in a higher incidence of major cardiovascular events, including myocardial infarction and acute heart failure [4]. The overall deterioration in health that occurs in post-sepsis patients might be due to progression of chronic disorders, residual organ insufficiency, and inadequate immune system functioning, and presumably a proper management of the acute phase of sepsis may result in less long-term complications [5].

Oxidative stress and inflammation are two central mechanisms that play a key role in the pathophysiology of sepsis. Therefore, attempts based on antioxidants and anti-inflammatory drugs have been tested in the therapeutic focus of this pathology. Thus, recent studies have been developed using antioxidant drugs, such as selenium [6], vitamin C [7], vitamin E [8], and N-acetylcysteine (NAC) [9], among other agents, for the treatment of sepsis. In addition, since uncontrolled inflammation would be exclusively responsible for the complications in patients with sepsis, several clinical trials aimed in the management of sepsis have been performed with drugs blocking the inflammatory cascade, including corticosteroids, anti-endotoxin antibodies, tumor necrosis factor and interleukin-1-receptor antagonists [10]. However, up to date, the results have not been consistent enough to recommend their use in sepsis guidelines [3]. In this review it is proposed to further explore the mechanism whereby oxidative stress becomes involved in the onset and progression of sepsis. Furthermore, due to the multifactorial factors contributing to the development of sepsis, we provide bases supporting the hypothesis that an increased efficacy against oxidative stress should be achieved by administering a variety of antioxidant agents. This combination may exert an additive or synergistic effect, potentially serving as a pharmacological adjunct to standard therapy to reduce sepsis complications.

## 2. Oxidative Stress and Sepsis 

### 2.1. Normal Response to Infection and Development of Sepsis

The normal host response to an infection begins with the recognition of microbial components by innate immune cells, leading to the release of proinflammatory cytokines such as interleukins, chemokines, and adhesion molecules, with the objective of recruiting additional inflammatory cells to eliminate the invading agents [11]. Under normal circumstances there is a balance between proinflammatory and anti-inflammatory mediators, resulting in a controlled inflammatory process, which enables the host to overcome the infection, leading to subsequent tissue repair. In contrast, when there is a dysregulation of inflammation, the phenomenon known as sepsis occurs [11]. The pathophysiological characteristics of sepsis involve a circle around a dysregulated response to infection, with an excessive release of proinflammatory cytokines [12], which are the mediators responsible for the systemic damage occurring under this condition [10]. 

### 2.2. Endothelial Dysfunction

The endothelium plays a fundamental role in vascular homeostasis as one of the main factors regulating the vasomotor tone [13], but it also constitutes a selective barrier to maintain tissue fluid homeostasis [14]. Moreover, in the presence of pathogens, the endothelium participates in preventing the spread of infection by releasing cytokines in order to recruit leukocytes and activate clot formation [13].

One of the main pathophysiological events in sepsis is endothelial dysfunction, which leads to a dysregulation of the vasomotor tone, a local imbalance between proinflammatory and anti-inflammatory species [12], and an increased permeability or loss of barrier functioning, with the latter resulting in a shift of circulating elements and tissue edema [14]. When there is a deep endothelial dysfunction a state of vasoplegia occurs, in which the endothelium loses the ability to regulate the vascular tone according to the metabolic demands of the tissues, thereby contributing to impaired microcirculatory blood flow and tissue hypoperfusion [15]. Accordingly, a state of hypoxia develops at the cellular level, leading to the activation of anaerobic metabolism with increased lactic acid production as a by-product [16], which is an independent mortality predictor in septic patients [17].

On the other hand, as part of the endothelial dysfunction a procoagulant state occurs, that causes development of intravascular fibrin deposition and clot formation, this phenomenon is called disseminated intravascular coagulation (DIC) [18]. It has been identified that the mechanism causing this event results from the inactivation of the metalloproteinase ADAMTS-13, which is the main inhibitor of the microthrombogenesis process [14]. Inflammatory mediators such as IL-6 and neutrophil-derived reactive oxygen species (ROS) are within the main agents responsible for the inactivation of this metalloproteinase [14]. In a later stage of DIC, bleeding may occur in parallel because of consumption of clotting factors and inhibitors [18]. 

### 2.3. Systemic Complications in Sepsis

On a systemic level, the major events that can be identified in patients with sepsis are myocardial dysfunction, sepsis-induced acute kidney failure (S-AKI), and impaired immune system functioning, leading to a late stage of immunosuppression [19] with increased susceptibility to new infections.

Myocardial dysfunction in septic patients is characterized by reversible biventricular dilation, decreased ejection fraction, and impaired response to fluid resuscitation and catecholamine stimulation [20]. Although mortality is substantially increased in patients with sepsis who develop cardiac dysfunction, these alterations appear to be reversible after 7–10 days in survivors [21]. This reversibility may be because myocardial damage in sepsis does not cause significant cell death in cardiac tissue, but a non-functioning hibernation-like state as part of an adaptive response to this phenomenon instead [10]. The foregoing is consistent with histological findings in patients who died of sepsis, where cell death in the heart was relatively minor and did not correlate with the profound level of organ dysfunction in these patients [10]. It is hypothesized that the depression of myocardial function occurs firstly due to mitochondrial dysfunction, which leads to excessive ROS production, and secondly because of a decreased nitric oxide (NO) availability, which is highly correlated with the contractile dysfunction observed in patients with sepsis [20].

Sepsis-induced AKI is a frequent complication in patients with sepsis, and it has been estimated that 1 out of 3 patients admitted for sepsis will develop AKI [22]. Originally, it was believed that S-AKI was caused by low renal blood flow, leading to renal tubular cell death; however, later studies showed that S-AKI could be present under conditions of normal or increased renal blood flow [23]. Likewise, it was found that histologically there was no significant tubular epithelial cell death (<5%), but heterogeneous, focal, and patchy tubular injury coupled with minor focal mesangial expansion instead [24]. It has been recognized that the activation of Toll-like-receptors (TLR) present in the plasma membrane of renal epithelial cells by bacterial elements activates the production of proinflammatory cytokines, leading to a local inflammatory phenomenon associated with mitochondrial injury and excessive ROS production [22]. The damage caused by these reactive species leads to a metabolic reprogramming of the renal cells to enter in a hibernation-like state, as previously mentioned in myocardial dysfunction due to sepsis, which prevents the death of these cells but compromises their function and thus affecting renal function [25].

Additionally, the gastrointestinal tract during sepsis is subjected to important changes. In particular, the most important ones being the intestinal barrier hyperpermeability, intestinal epithelial apoptosis and dysbiosis [26]. Indeed, during sepsis, the physiological integrity of the intestine, achieved through a balance of anti-inflammatory and inflammatory responses, is compromised [26]. This results in a disruption of the equilibrium between the host and its bacterial colonizers, leading to bacterial translocation, which, in addition to the hyperpermeability of the intestine, allows the passage of these pathogens into the systemic circulation [27]. Therefore, the intestine plays a very important role both in the worsening of intra-abdominal sepsis and in the spread of the infectious phenomenon to the rest of the body.

Although the different complications in sepsis result from a deregulated pro-inflammatory response, after two hours of the sepsis onset an immunosuppressive phenomenon begins to be triggered [28] due to the depletion of CD4 and CD8 lymphocytes because of their splenic sequestration and also apoptosis of CD4 lymphocytes induced by the large number of cytokines in the early stage of sepsis development [28]. Therefore, the late phase of sepsis is characterized by an increased risk of secondary infections [28]. This correlates with postmortem findings in patients who died of sepsis, in whom a depletion of T cell populations was found compared to patients who died of non-infectious causes [29].

As identified in the previous sections, OS is transversally present in the pathophysiology of sepsis and its complications, so understanding this phenomenon is crucial for the identification of therapeutic agents able to target this phenomenon.

### 2.4. Oxidative Stress Definition and ROS Sources

Oxidative stress is a phenomenon that occurs because of an imbalance between oxidant potential and antioxidant defense system activity, in favor of the oxidants, leading to a disruption of the redox cell signaling system associated with molecular damage [30].

There are reactive species derived from oxygen, nitrogen, and sulfur, with the first two being particularly relevant in the pathophysiology of sepsis. Among the ROS, the superoxide radical anion (O_2_**^−^**), hydrogen peroxide (H_2_O_2_), hydroxyl radical (•OH), and oxygen singlet (^1^O_2_) are the most important. On the other hand, reactive nitrogen species (RNS) include NO**˙**, peroxynitrite anion (ONOO**^−^**), and nitrogen dioxide radical (NO_2_) [31], among other species. It is important to emphasize that ROS and RNS are involved in a multitude of biological processes, but when found at specific concentrations and sites, over a specific time span, they are not associated with cellular damage [30]. Indeed, the first function of ROS that has been discovered is their role in neutrophils phagocytosis. What happens is that upon phagocytosis of pathogens, the ROS are produced by the NADPH oxidases (NOX) in the small volume of the phagosome [32].

Another example of the physiological action of ROS is endothelium-dependent tone homeostasis, regulated by ROS generated on the mitochondria. On exposure to cardiovascular risk factors, endothelial mitochondria produce excessive ROS in concert with other cellular ROS sources. Mitochondrial ROS, in this setting, act as important cell signaling molecules, activating prothrombotic and proinflammatory pathways in the vascular endothelium [33].

The major ROS sources include NOX, uncoupled endothelial nitric oxide synthase (eNOS), xanthine oxidases (XO), and the electron transport chain at the mitochondrial level [34]. NOX catalyzes the formation of O_2_**^−^** using NADPH as an electron source and is extensively present in activated neutrophils, in which O_2_**^−^**, together with other ROS, has a bactericidal function [34]. Endothelial nitric oxide synthase plays a fundamental role in the regulation of vascular tone. Endothelial nitric oxide synthase uncoupled in the absence of one of its substrates L-arginine or the cofactor tetrahydrobiopterin (BH4), resulting in the formation of O_2_**^−^** instead of NO [34]. Xanthine oxidase is normally found to be present in the vascular endothelium, participating in uric acid formation with O_2_**^−^** co-production [31]. In sepsis, activation of TLR4 induced by lipopolysaccharide (LPS) or TLR7 and TLR8 by (+) ssRNA virus leads to activation of XO, with consequent formation of large amounts of O_2_**^−^** [35]. Superoxide leads to the recruitment of neutrophils and their adhesion to endothelial cells, which stimulates the formation of XO in the endothelium and thus increased superoxide formation [36]. The mitochondrion is involved in energy generation processes in the cell and is likewise an organelle that is a source of ROS [37]. When there is mitochondrial damage, the electron transport chain is unable to function properly leading to the formation of large amounts of ROS, which causes damage to the mitochondrial DNA and consequent increased ROS formation [37]. Additionally, damage to the mitochondrial membrane, either by external sources or by the generation of ROS within the mitochondrion itself, results in cytochrome c release, caspase activation, and apoptosis [38].

### 2.5. Antioxidant Defense System

On the other hand, there is the antioxidant defense system, which has enzymatic and non-enzymatic mechanisms to counteract the production of ROS [31]. Enzymatic mechanisms include superoxide dismutase (SOD) that converts O_2_**^−^** to O_2_ or the less reactive H_2_O_2_, glutathione peroxidase (GSH-Px) that catalyzes the conversion of H_2_O_2_, to water by converting reduced glutathione (GSH) to oxidized glutathione (GSSG) and catalase (CAT) that also catalyzes the breakdown of H_2_O_2_, [39], all of which constitute the first line of defense against the overproduction of ROS. Non-enzymatic mechanisms include a vast array of biomolecules such as vitamin C, α-tocopherol (vitamin E), GSH, carotenoids, flavonoids, polyphenols, and other exogenous antioxidants [40].

The role of SOD as part of the antioxidant defense system is reflected in a study in rats in which supplementation with artificial SOD (Glisodin^®^) led to lower levels of the RNS nitrosamine and proinflammatory cytokines, which resulted in a lower incidence of S-AKI in the SOD-supplemented group [41]. GSH-Px is a selenoprotein, therefore adequate levels of selenium are required for the proper functioning of this enzyme [39]. Consequently, a correlation has been observed between low selenium levels and greater severity in patients hospitalized for sepsis [42]. Moreover, it has been observed that Glutathione peroxidase 3 (GSH-Px-3) activity is decreased in patients with sepsis, due to a lower expression of GSH-Px-3, as these patients usually have lower levels of selenium, which limits the synthesis of this enzyme [43].

The development of OS occurs in the context in which the production of ROS exceeds the capacity of the antioxidant defense system, resulting in damage to different biomolecules [30]. Protein carbonylation occurs by oxidative cleavage of protein backbones, resulting in structural modifications that affect the proper functioning of the protein [44]. At the membrane level, ROS generates lipid peroxidation (LPO), which alters the integrity of the cell membrane and membranous organelles [44]. Finally, DNA damage can also be found mainly in the guanine nucleotides, where oxidative modifications to the DNA result in mutations [44].

## 3. Antioxidant Treatments in Sepsis

Considering the role of OS in the pathophysiology of sepsis, the following section will explore a few antioxidant agents that have proven to have a beneficial effect against the deleterious effects of sepsis. The studied drugs are vitamin C, selenium, NAC, and vitamin E.

### 3.1. Vitamin C

Vitamin C, or ascorbate, is a water-soluble antioxidant compound that forms part of the antioxidant system in humans. Its antioxidant capacity comes from being an electron donor, reducing free radicals, and being oxidized to dehydroascorbate [31]. Normal levels of vitamin C vary between 50 to 70 µmol/L. The most described vitamin C transporters are Na^+^-dependent cotransporters SVCT1 and SVCT2 [31]. Many of the physiologic roles of vitamin C are important in patients with sepsis. These include the key antioxidant properties of vitamin C as ROS scavenging, stabilization of eNOS activity, repletion of other crucial body antioxidants such as vitamin E and GSH [31], and increase of norepinephrine and vasopressin endogenous synthesis. It acts as a cofactor of dopamine β-hydroxylase and tyrosine hydroxylase in the synthesis of norepinephrine [45]. Likewise, vitamin C acts as a cofactor in the synthesis of L-carnitine, which can reduce TNFα production, thus decreasing the severity of septic shock [46]. However, vitamin C not only acts as a direct antioxidant, but studies in rats have also shown that it has an inhibitory effect on NOX and iNOS, enzymes that are major sources of ROS [47].

In critically ill septic patients, plasma vitamin C levels greatly decrease and facilitate ROS and RNS generation [48], which is consistent with findings that correlate low vitamin C levels to a higher incidence of organic failure and worse outcomes on septic patients [49]. A clinical trial showed that 28-day mortality was significantly lower in septic patients receiving vitamin C, with a significant increase in intensive care units (ICU)-free days up to day 28, and hospital-free days up to day 60. However, these results were based on analyses that did not consider multiple comparisons and should therefore be considered exploratory [6]. It has been reported that vitamin C reduces 28-day mortality and dosage and duration of norepinephrine in intermittent intervals [50], also reducing the risk of pulmonary morbidity and organ failure [51]. Lamontagne et al. reported no statistical differences between placebo and supplemented group [50].

### 3.2. Selenium

Selenium is an essential micronutrient that acts as an enzymatic cofactor of more than 30 selenoproteins [52]. This protein group has different biological functions, particularly related to redox cell signaling and antioxidant response, thyroid hormone metabolism, and humoral and cellular immune response [53].

Approximately 60% of serum selenium is incorporated to selenoprotein P (SePP), 30% to GSH-Px, and 5–10% to albumin [52]. The GSH-Px family catalyzes various hydroperoxides reductions and has synergy with vitamin E in antioxidant defense against LPO [54]. The thioredoxin reductase family catalyzes the conversion of H_2_O_2_ to H_2_O. Endothelial dysfunction caused by sepsis leads to SePP adhesion to the endothelium, which is believed to be a protection mechanism against major damage caused by OS [52]. In catabolic states, selenium urinary excretion increases. It has been described that selenoproteins inhibit NF-κB by redox cell signaling and therefore reduce cytokine storm and ROS/RNS production [55].

Critically ill patients coursing sepsis have lower levels of selenium [52], and concentrations lower than 0.7 µmol/L have been associated with higher mortality and organ dysfunction in patients on ICU [6].

In a clinical trial developed by Angstwurm et al. [56], in patients with severe sepsis or septic shock and APACHE III score > 70 who were infused with 1 mg of sodium selenite followed by continuous 1 mg daily infusion for 14 days, a decrease was shown in 28-day mortality versus placebo group, and adequate selenium and GSH-Px-3 serum levels were found. Adjuvant therapy with a continuous dose of 750 μg/24 h of sodium selenite could be beneficial in septic patients with acute lung injury, as shown by Kočan et al. [57].

Evidence surrounding selenium treatment benefits for sepsis as a monotherapy is controversial. In a metanalysis made by Kong et al. [58], selenium administration as an adjuvant to standard therapy in severe sepsis of septic shock showed no difference in 28-day mortality but showed a decrease in all-cause mortality.

### 3.3. N-acetylcysteine

N-acetylcysteine is a thiol precursor of L-cysteine, which is the rate-limiting amino acid for GSH synthesis [59]. L-cysteine is transported to the intracellular space by the Na^+^-dependent alanine–serine–cysteine system. Nevertheless, because of NAC’s permeability through the membrane, it does not require active transport [60]. Between the three amino acids which GSH is composed of (glutamate, glycine, and cysteine), cysteine is the least concentrated in the cytosol [60], therefore in OS, its concentration determines the rate of GSH synthesis, and, in consequence, GSH-Px activity as part of the antioxidant response. Additionally, NAC acts as a direct ROS scavenger, as it can be oxidized by multiple free radicals, and it reacts with H_2_O_2_, resulting in H_2_O as a byproduct [61]. Given NAC’s important role as GSH precursor and its direct action in ROS and RNS, multiple clinical trials have been done where NAC supplementation could be beneficial in pathologies where OS has a major role in their development.

In a study developed by Spies et al. [9], 58 patients that required hemodynamic support who developed sepsis were given 150 mg/Kg of NAC in 15 min followed by lowering it to 12.5 mg/kg for 90 min. Forty-five percent of the patients responded by an increase of >10% of VO_2_, and a decrease in PCO_2_ levels, concluding an increase in tissue oxygenation and cardiac function.

### 3.4. Vitamin E

α-tocopherol, the main molecular form of vitamin E, is a lipid-soluble molecule capable of regulating the production of reactive species in mitochondria in a dose-dependent manner [62]. Vitamin E acts as a free-radical scavenging antioxidant, however it does not act against non-radical oxidant species [63].

A study made by Weber et al. [64] reported an association between low selenium levels and α-tocopherol independently with higher levels of apoptosis in patients with severe sepsis versus non sepsis ICU patients and healthy patients.

In septic shock patients, vitamin E is associated with decreased levels of procalcitonin [7].

A study made with pig models showed an indirect association between vitamin E levels and LPO byproducts, and a rapid increase in oxidative stress biomarkers [65].

## 4. Discussion

Considering the multifactorial characteristics of the pathophysiological mechanisms being exerted in sepsis, in this study we put forward the hypothesis of an enhanced efficacy of a potential antioxidant multitherapy against complications occurring in this pathology. Sepsis is a disproportionated response to an infection [3]. Normally, what happens during an infection is a controlled inflammatory process occurring through a physiological balance between inflammatory and anti-inflammatory mediators [3]. When sepsis is not controlled, the cascade of events that follows can progress to septic shock and multi organ failure [2]. Among the various pathophysiological mechanisms accounting for this transition, oxidative stress has a key role [10].

It is noteworthy that during oxidative stress in sepsis there are three main mechanisms involved: vasomotor impairment, necrosis in different tissues, and mitochondrial dysfunction [13,15]. Thus, an excessive presence of ROS leads to a disruption of the physiological endothelial function, causing vasoplegia, an excessive procoagulant state caused by the inhibition of ADAMTS-13 enzyme [12], and damage to the mitochondria, disrupting the signaling pathways that regulate thrombosis and inflammation [14].

There are some antioxidant molecules that can target all these reported mechanisms. In this paper we specifically focused on vitamin C, vitamin E, NAC, and selenium. Vitamin C and Vitamin E are ROS scavenger molecules [48,62]. Vitamin C can act both on the directly by scavenging ROS and indirectly abrogating ROS production. In turn, vitamin E can regulate mitochondrial ROS overproduction and both vitamins can contribute to counteract the development of endothelial dysfunction [48,62]. The role of selenium is related to the modulation of the activity of antioxidant enzymes, since it participates in the synthesis of GSH-Px [52], whose action is further enhanced by NAC as donor of GSH [61], also leading to a reduction of the endothelial dysfunction. Furthermore, NAC is also a direct ROS scavenger [61], thus reducing oxidative stress in a dual manner. Data here presented contribute to support our hypothesis that an antioxidant multitherapy, mainly due to an additive or synergistic antioxidant effect of all the molecules, can be an important adjunctive therapy to sepsis current treatment.

There are already studies demonstrating that the use of an antioxidant therapy can tone down the inflammation, which leads to sepsis. Aisa-Alvarez et al. [8] showed a reduced inflammation in patients with septic shock, achieved by adding several antioxidants to standard therapy. In this study, the antioxidants employed are vitamin C, NAC, and vitamin E. In pulmonary sepsis, replacement therapy with vitamin C decreases the levels of CRP, PCT, and NO_3_^−^/NO_2_^−^, NAC reduces LPO and improves the antioxidant capacity, and vitamin E tends to decrease LPO. In summary, each antioxidant had contributed with a beneficial effect [8].

Nevertheless, there are studies concerning antioxidant therapies where some side effects are shown. In particular, Dodd et al. studied the pharmacology and clinical utility of NAC, highlighting that the main side effects of NAC therapy are mild gastrointestinal symptoms [66]. Indeed, side effects of NAC therapy are predominantly benign, and more serious side effects are dose-dependent. If the intravenously administered NAC dosage is >3 g/day, anaphylactoid reactions, including flushing, urticaria, bronchospasm, hypo-tension, and angioedema, can occur [67].

However, other negative effects regarding antioxidant monotherapies have been brought up, in particular concerning the cardiovascular and renal systems. With respect to the renal system, it seems that vitamin C is associated with increases in AKI and in-hospital mortality [68,69] and an early administration of NAC not only does not attenuate the endothelial damage during sepsis, but it can aggravate it, leading to sepsis-induced organ failure and, in particular, cardiovascular failure [70]. Moreover, it seems that NAC administration can directly damage the cardiovascular system, leading to cardiac depression, reduction of cardiac index, left ventricular stroke work index, and mean arterial pressure [67]. Additionally, vitamin C can have more systemic side effects such as nausea and vomiting, fatigue, irritability, and coagulation problems [71].

On the other hand, most studies use supraphysiological doses of vitamin C in order to reduce complications in sepsis. High doses of vitamin C have been associated with an increased conversion of this compound to calcium oxalate in a dose-dependent manner, which could lead to calcium oxalate nephropathy [46].

To date, studies concerning an antioxidant monotherapy in septic patients have found controversial results. The study of Woth et al. [72] showed that Na-selenite usage in septic patients with multiple organ failure did not improve white blood cell antigen expression. In a clinical trial developed by Bloos et al. [73], patients with severe sepsis or septic shock that were infused with 1 mg of sodium selenite followed by continuous 1 mg daily infusion until discharge showed no significant difference in 28-day mortality versus placebo group. In addition, several studies reported that vitamin C and selenium used as monotherapy did not appear to have any effect in terms of reducing sepsis complications and mortality. Vitamin C used alone as monotherapy has shown controversial results in patients with sepsis, where no significant differences in markers of organ dysfunction have been observed [7]. Indeed, other clinical trials have suggested an increased incidence of sepsis-induced AKI [68] and increased risk of mortality [74]. In the case of selenium, clinical studies have not shown that this antioxidant has a significant effect in reducing mortality in patients with sepsis [71,72,75], despite showing a substantial increase in the GSH-Px enzyme as part of the antioxidant defense system [71,75,76].

Table 1 describes the clinical studies presented in this section of this review in more detail.

Given the lack of significant benefit in clinical studies with the use of antioxidants monotherapy for the management of sepsis, we suggest the use of multitherapy. Indeed, given the complexity of the condition, the multiple mechanisms that are involved in the development of the pathology and the heterogeneous group of patients that develop the condition. It is of interest to use therapies that are complemented by additively or synergistically acting effects on different pathways that contribute to a reinforcement of the antioxidant defense [77], which can be done by administering a variety of antioxidant agents, as shown in Figure 1. This could have a significant and optimal effect as a novel management, likely contributing to reduce the mortality of this pathology as well, as was suggested for other complex diseases that have different pathways as reperfusion injury [78].

In agreement with this view, there is evidence in animal models demonstrating the potential benefit of antioxidant multitherapy in preventing complications in sepsis. Carlson et al. examined the effects of the combination of vitamin C, vitamin E, vitamin A, and Zinc on several aspects of sepsis-related myocardial signaling cascades, in which the rats in the group that received the intervention presented attenuated myocardial inflammatory cytokine signaling and diminished sepsis-related contractile dysfunction [79]. Moreover, Galvão et al. performed a study in which a combination of ceftriaxone, together with NAC and vitamin C and E, was administered after the induction of sepsis in rats. Survival in the ceftriaxone plus antioxidants group was 83.2% compared to 66.6% in the control group in which ceftriaxone alone was administered [80]. Additionally, in this study it was found that the administration of antioxidants was associated with a decrease in superoxide radical anion, lipid peroxidation, and protein carbonylation levels [80], all of which are markers of oxidative stress.

On the other hand, there are not many studies testing a combination of antioxidants for the treatment of sepsis in humans, with only a few being relevant. Among these, Galley et al. [81] conducted a randomized clinical trial where 30 patients with septic shock were randomized in two groups. One group was given 150 mg/kg over 30 min of NAC, then 20 mg/kg/h plus boluses of NAC, 1 g of vitamin C, and 400 mg of vitamin E. The other group was given a placebo. Antioxidant administration was associated with beneficial hemodynamic changes. This study confirmed the role of free radicals in the inflammatory process and suggested an alternative therapeutic strategy for the treatment of the hemodynamic derangements of septic shock.

Having observed that the studies regarding antioxidant monotherapy either reported a low significance or described several side effects, it is clear from published literature that the most effective form of antioxidant repletion is likely to consist in combinations of antioxidants with known synergistic actions [81]. We suggest further research on the efficacy of a complementary antioxidant multitherapy to reduce the incidence of organic failure and thus ameliorate the clinical outcome of septic patients.

## Figures and Tables

**Figure 1 biomedicines-10-03088-f001:**
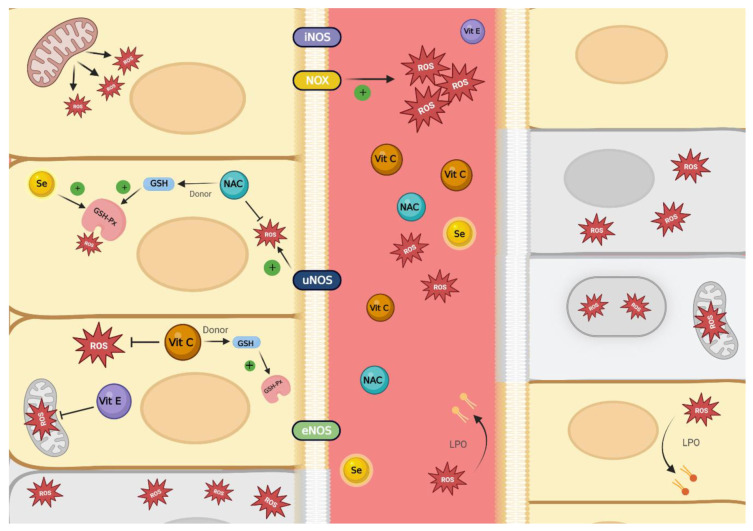
Schematization of oxidative stress mediated endothelial and mitochondrial dysfunction in sepsis pathogenesis, and antioxidant protection. eNOS: endothelial nitric oxide synthase; GSH: glutathione; GSH-Px: glutathione peroxidase; iNOS: induced nitric oxide synthase; LPO: lipid peroxidation; NAC: n-acetylcysteine; NOX: NADPH oxidase; ROS: reactive oxygen species; Se: selenium; uNOS: uncoupled nitric oxide synthase; Vit C: vitamin C; Vit E: vitamin E.

**Table 1 biomedicines-10-03088-t001:** Further details on clinical studies carried out with regard to antioxidant monotherapy.

Antioxidant	Study Details	*n*		Main Findings	Ref.
		Intervention Control			
Ascorbate	Vitamin C intravenous infusion in patients with sepsis and ARDS for less than 24 h.	84	83	There were no significant differences in organ dysfunction scores or markers in inflammation and vascular injury between both groups.	[7]
	Intravenous vitamin C in patients with sepsis receiving vasopressor therapy.	435	437	The risk of death or persistent organ dysfunction at 28 days was higher in the Vitamin C group.	[74]
	Effect of high-dose ascorbic acid on vasopressor drug requirement in surgical patients with septic shock.	14	14	The mean dose of epinephrine and duration of norepinephrine administration was significantly lower in the ascorbic acid group.	[50]
	Effects of vitamin C therapy on acute kidney injury (AKI) and mortality among septic patients.	212	1178	The occurrence of AKI in ICU patients was significantly higher in the vitamin C group, with no protective benefit against mortality.	[68]
	Effect of Vitamin C on mortality of critically ill patients with severe pneumonia in intensive care unit.	40	40	Duration of mechanical ventilation and vasopressor use were significantly lower in the vitamin C group.	[69]
	Effect of orally administered vitamin C adjuvant treatment on septic patients with multi organ failure.	18	21	The vitamin C group had decreased C reactive protein levels and the nitrate/nitrite ratio.	[8]
NAC	Continuous N-acetylcysteine infusion adjuvant to standard sepsis therapy.	10	10	NAC infusion was associated with a depression in cardiovascular performance. No differences in hospital mortality rate.	[67]
	Continuous N-acetylcysteine infusion adjuvant to standard sepsis therapy.	18	17	Worsening of organ failure and particularly cardiovascular failure in the NAC treatment group.	[70]
	Effect of orally administered N-acetylcysteine adjuvant treatment on septic patients with multi organ failure.	20	21	The NAC administered group had a decrease in procalcitonin levels and increased antioxidant capacity.	[8]
Selenium	Sodium-selenite in patients with severe sepsis to reduce oxidative stress.	21	19	Serum selenium levels increased, and ROS decreased significantly. However, there was no significant difference in the survival rate and SOFA score between both groups.	[72]
	High-dose intravenous administration of sodium-selenite in patients with severe sepsis or septic shock.	543	546	There was no statistical difference in the 28-day mortality between both groups.	[73]
	High-dose supplementation of sodium-selenite as an adjuvant therapy in patients with severe sepsis or septic shock.	92	97	The adjuvant therapy of sodium-selenite reduced the mortality rate.	[56]
	High dose compared with standard dose (control group) of selenium in patients with SIRS/sepsis with SOFA >5.	75	75	High dose selenium substitution increased serum levels of selenium and GSH-Px levels but did not reduce mortality.	[71]
	Parenteral selenium in mechanically ventilated patients with sepsis or septic shock.	29	25	The 28-day mortality was not significantly different. The activity of GSH-Px increased. Additionally, selenium supplementation was associated with reduced occurrence of ventilator-associated pneumonia.	[75]
	Immune function testing in sepsis patients receiving sodium selenite	40	36	An overall dampening in cytokine release was observed in the selenium group.	[76]
Vitamin E	Effect of orally administered vitamin E adjuvant treatment on septic patients with multi organ failure.	18	21	The group supplemented with vitamin E had decreased levels of procalcitonin.	[8]

## Data Availability

Not applicable.

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
