# Peer review of "Potential Antioxidant Multitherapy against Complications Occurring in Sepsis"

_biomedicines, 2022, doi:10.3390/biomedicines10123088_

Round 1
Reviewer 1 Report
General comments
The MS No.: Manuscript ID_ biomedicines-1975283: ”Potential antioxidant multitherapy against complications occurring in sepsis” by authors: Joaquin Abelli, Gabriel Méndez-Valdés, Francisca
Gómez, María Chiara Bragato, Silvia Chichiarelli, Luciano Saso and Ramón Rodrigo, represent a review article considering the use of combined antioxidant therapies in order to improve clinical outcome of patients with sepsis or septic shock, in the view of significance of oxidative stress in the pathogenesis of multi-organ dysfunction in sepsis. Particular attention is paid to the use of antioxidants such as ascorbate, N-acetylcysteine and selenium. Authors hypothesize oxidative stress is responsible in this multi-organ disease and that a supplementation of more than one antioxidant compound causes their synergistic effect leading to better prognosis in patients with sepsis or septic shock.
The title of MS is clear and adequate.
Abstract is well written, clear and and self-explanatory for the readers.
The introduction section is well written and explains the basic concepts of sepsid from cellular to systemic damages with the special emphasis on oxidative stress caused by sepsis. Authors potentiates the influence of endothelial disfuncion in the sepsis pathogenesis and the role of antioxidant supplementation on improvement of prognosis of this disease. Special attention is focused on the impact of the combination of antioxidant supplements and concluded that antioxidants such as Vit C, N-acetylcysteine, Vit E and Se have a synergisic influence on systemic complications of sepsis.
Sepsis as a very insidious and often fatal condition has been extensively researched, but so far no single formula has been found that leads to the recovery of most patients. That is why this review paper represents a contribution to the body of knowledge in the treatment of this disease.
Materials and methods
Finally, the review represents a contribution to the overall scientific knowledge in this area and provides a solid basis for further investigations.
I have no specific comment in the text.
Having all the above in mind, I suggest to the editor to accept this review manuscript for publication in the present form.
My final opinion: acceptable for publication.
Author Response
The authors are thankful for your comments
Reviewer 2 Report
The authors describe an interesting argument for using antioxidant therapy for better outcomes in patients with sepsis. The articles reviewed in this manuscript describe both beneficial as well as non-significant impact of vitamin C, selenium, NAC, and vitamin E. However, limited evidence is presented to support multi-therapy in sepsis. More thorough examination of multi-therapy is needed. Alternatively, the title should be changed to more directly reflect the details in this review article.
Author Response
The authors thank you for your comments. We take into account that in the mentioned subsection there is limited evidence about multitherapy, so we conducted a new search, from which we were able to rescue studies done in animal models, since clinical studies of combination of antioxidant drugs are scarce.
Author Response
Greetings, the document with the response to reviewer 3 comments are attached to this message. Thank you for your time.

Round 2
Reviewer 2 Report
I support the publication of this manuscript.
Author Response
We are thankful for your comment
Author Response
We are grateful for your observations and we changed the text according to your suggestions.